# Research on Vegetation Coverage Dynamics and Prediction in the Taitema Lake Region

**Tingfang Zuo** [1,2,3], **Yaning Chen** [2,*] **and Jianli Ding** [1]

1   Xinjiang Common University Key Laboratory of Oasis Ecology of Ministry of Education, College of Resource and Environment Sciences, Xinjiang University, Urumqi 830046, China; zsmlm13002@hotmail.com (T.Z.); watarid@xju.edu.cn (J.D.)

2   State Key Laboratory of Desert and Oasis Ecology, Xinjiang Institute of Ecology and Geography, Chinese Academy of Sciences, Urumqi 830011, China

3   College of Resources and Environment, University of Chinese Academy of Sciences, Beijing 100049, China

\*   Correspondence: chenyn@ms.xjb.ac.cn; Tel.: +86-99-1782-3169

**Abstract:** The Tarim River is the largest inland river in China, which plays a crucial role in maintaining regional ecological security and carbon cycle/dynamic. However, the "green corridor" in the Taitema Lake region at the lower reaches of the Tarim River has unclear environmental changes and future dynamics due to the influence of the ecological water conveyance. Hence, protecting the "green corridor" at the lower reaches of the Tarim River in China is strategically important not only ecologically but also socially and economically. In this paper, the temporal and spatial features of the fractional vegetation coverage (FVC) dynamics in the Taitema Lake region at the lower reaches of the Tarim River in 2000–2018 are analyzed and calculated using Landsat TM/OLI remote sensing images and MODIS data products. Additionally, the future trend of FVC dynamics in the study region are predicted using trend analysis and the pixel-based Hurst index. The results show that FVC in the Taitema Lake region exhibit a positive development after the implementation of ecological water conveyance. Specifically, from 2000 to 2018, the areas of low, medium, and high FVC expanded from 1.28 km$^2$ to 179.87 km$^2$, resulting in an increase of 140.52%. Spatially, the regions around the lake entrance channel of the Tarim River saw a significant increase in FVC of 9.71%. The middle part of the study region, accounting for only 1.96% of the area, displayed relatively high and high fluctuations in FVC. In the future, the regions at the middle part of the lake and around the lake entrance channel of the Tarim River, accounting for 11.33% of the area, will likely show an increasing trend in FVC. The regions with either extremely low or low FVC are predicted to decrease to 14.16% of the overall area. Because the positive effects of ecological water conveyance were more significant on FVC in the study region than the influences of either temperature or precipitation, ecological water conveyance should remain the primary means of ecological restoration for Taitema Lake.

**Keywords:** ecological water conveyance; FVC; hurst index; future trend; Taitema Lake



## 1. Introduction

As vital water resources in arid regions, terminal lakes of inland river basins can maintain the positive cycle of the ecosystem, improve local climate, restrain sandstorms, and maintain biodiversity [1,2]. The eco-hydrological effect of such lakes is generally expressed as the correlation between vegetation and moisture characterized by natural precipitation, surface water, and groundwater [3]. Large amount of surface water at the upper and middle reaches of inland river basins is consumed due to human activities, exacerbating water-use conflicts at the lower reaches. Besides, the groundwater at the lower reaches is overexploited, leading to reduced vegetation and reactivation of fixed dunes, and severely affecting the ecological stability of desert oases. To restore the ecosystem of artificial vegetation and natural vegetation on the edges of deserts and oases, replenishment of moisture for vegetation growth through ecological water conveyance to downstream rivers

is an emergency measure for vegetation restoration at many inland rivers, including Tarim River and Heihe River. According to various studies [4–6], ecological water conveyance is significantly effective for ecological restoration of inland rivers, implying that desert vegetation and moisture are interdependent; that is, moisture affects vegetation growth and distribution. Vegetation restoration in many basins has been widely assessed [5,6], but the emphasis has been placed on the effect of climatic variation on ecosystems in related studies. Such a restoration not only changes the eco-hydrological processes but also affects water interactions on a scale of the whole basin [7]. In particular, there are changes in groundwater and soil water environments after artificial water conveyance. Hence, for vegetation restoration at terminal lakes at present, it is urgent to find out the response of the original natural vegetation at terminal lakes to the rapidly changing water environment.

The "green corridor" at the lower reaches of the Tarim River is located in the eastern part of the Tarim Basin in Xinjiang, China, between the Taklimakan and Kuruk deserts. Topographically, this region is high in the north and low in the south and high in the west and low in the east, with a curved channel. The elevation ranges between 801.50 and 846.25 m above sea level, with the Taitema Lake region as the lowest point (801.50 m).The lake is situated at the lower reaches of the Tarim River. As a terminal lake of the inland Tarim River in an arid region, Taitema Lake is of great significance for maintaining the stability of the regional ecosystem, conserving soil and water, preventing desertification, and protecting the safety of oases [8].

Since the 1960s, the water resources in the Tarim River Basin have been excessively developed along with the economy and population in southern Xinjiang. As a result, the downstream water volume has declined [9,10]. After the construction of the Daxihaizi Reservoir in the 1970s, 321 km of the downstream river channel was set off, Taitema Lake at the lower reaches of the river dried up, and the vegetation along the banks of the river suffered a massive die-off, resulting in decreased fractional vegetation coverage (FVC) and intensified desertification. Consequently, the ecological environment has deteriorated sharply, and the region has become the most deteriorated in western China [3,4,11].

The worsening ecological environment in the lower reaches of the Tarim River has not only attracted the attention of many scholars and the Chinese government but has also raised environmental concerns about other inland rivers, including the Heihe River and the Shiyang River in arid regions of western China. To halt the continuous deterioration of the ecological environment in inland river basins in arid regions, the Chinese government launched a series of unified management projects for inland rivers located in these regions [12]. Among them, a comprehensive management project for the Tarim River Basin was approved by the State Council and implemented in 2001 [13]. It includes the ecological water conveyance project (EWCP) that started conveying ecological water to the lower reaches of the Tarim River in 2000. In this process, the water is directly conveyed to the lower reaches of the river to replenish the groundwater around the river channel, rendering favorable conditions for the growth and restoration of endangered vegetation. The ecological water conveyed into Taitema Lake at the lower reaches of the Tarim River from the Daxihaizi Reservoir has brought water back to the lake, which had been dried up for 40 years. It has also greatly improved the balance of the regional natural ecological environment [14].

Combined with the study of ecological water conveyance, the ecological effects of inland river basins in arid regions [14,15] and other research areas have been evaluated from various perspectives by numerous scholars. For example, Chen et al. [14] analyzed several ecological indicators and concluded that the ecological environment in the lower reaches of the Tarim River is recovering, and the increases in vegetation area and FVC are larger in the middle reaches (Yingsu–Arakan) than in the upper (Daxihaizi Reservoir–Yingsu) and lower reaches (Arakan–Taitema Lake) of the river. In addition, the normalized difference vegetation index (NDVI) and FVC have large values in the areas within 2 km from the river channel and significant increases during water conveyance but then decline in the areas beyond 2 km. HAO et al. [16] determined the response of vegetation to groundwater in the

lower reaches of the Tarim River by investigating FVC dynamics characteristics of different plants in the lower reaches of the river based on long time-series Moderate Resolution Imaging Spectroradiometer (MODIS) data. A few years later, Bao et al. [17] uncovered the growth and restoration of vegetation in the lower reaches of the Tarim River by exploring the characteristics of FVC and land-use dynamics with long time-series MODIS data. These researchers discovered that FVC is incremental.

The studies on the middle and lower reaches of the Tarim River over different time periods show varying results, which refer to the characteristics of water and salt transport in desert riparian forests [18], the response of groundwater levels [19], the characteristics of spatial changes in surface water bodies [20,21], water-use efficiency [22], and ecological environment response [16]. Besides, the vegetation dynamic in the Taitema Lake region remains unclear, especially its future vegetation dynamic with the implementation of the ecological water conveyance project in the region. Therefore, in this study, Landsat and MODIS satellite data are employed to analyze FVC in the Taitema Lake region through trend analysis and the pixel-based Hurst index in order to determine the characteristics of FVC dynamics in the region from 2000 to 2018 and to predict future trends. The study findings are expected to offer a scientific basis for ecological protection and regional security.

## 2. Study Area and Analytical Method

### 2.1. Study Area

Taitema Lake, which has been selected as the study region, is the terminal lake of the Tarim River and Qarqan rivers (Figure 1). For forty years starting in the 1970s, Taitema Lake was in a dry state due to natural and human factors [14]. As a result, the region has become one of the most important, sensitive, and vulnerable places in the Tarim River Basin. In 2000, its water level began to be restored after the government implemented a comprehensive management project for the Tarim River Basin (including ecological water conveyance to the lower reaches of the river), providing an ecological environment for recovering FVC [23,24].

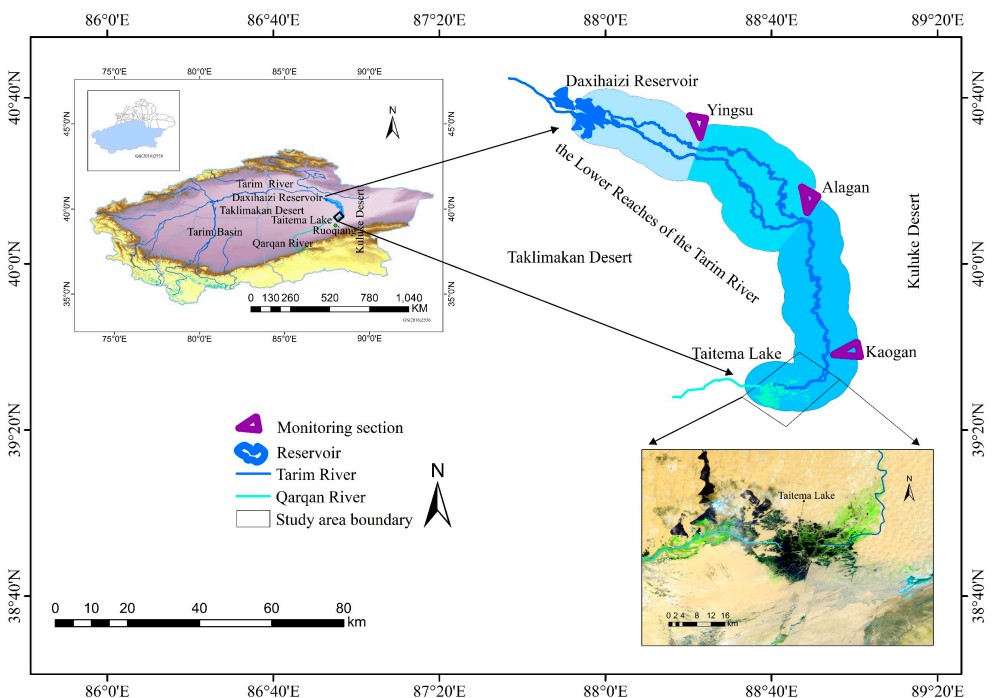

**Figure 1.** Geographical location of study area.

The Taitema Lake region has an extreme continental climate, making it extremely arid. The Ruoqiang meteorological station records show that the average annual precipitation

in the region is 17.4–42 mm, the average annual evaporation is 2500–3000 mm, and the extreme maximum temperature is 43.6 °C. Furthermore, above-ground vegetation is sparse, with natural vegetation mainly comprising *Tamarix ramosissima Lcdcb.*, *Populus euphratica*, *Haloxylon ammodendron* (*C. A. Mey.*) *Bunge*, *Alhagi sparsifolia Shap.*, *Tamarix ramosissima*, *Lycium ruthenicum*, *Apocynum venetum*, *Karelinia caspia*, and *Phragmites australis* [14].

### 2.2. Data Source

In this study, the data were mainly selected from Landsat TM/OLI remote sensing images with a spatial resolution of 30 m and a strip number of 141/033 for the four periods of 2001, 2007, 2014, and 2018. The data were downloaded from the United States Geological Survey (USGS) website. The study also used MOD13Q1 annual average NDVI data products with a spatial resolution of 250 m for 19 periods from 2000 to 2018 from NASA LAADS. In the selected Landsat images, the study region was not covered by cloud or fog, and the shots were taken during the growing season (June to September) so as to reduce any errors caused by seasonal differences. The images were subjected to pre-processing (radiation calibration, atmospheric correction, and geometric correction) by ENVI5.3 software, after which MRT software was used to project and convert the format of the MOD13Q1 data products. Following masking, the Landsat images and MOD13Q1 vegetation index data for the study region were finally obtained.

The meteorological data came from the China Meteorological Data Service Centre, with data on temperature, precipitation, and humidity obtained from the Ruoqiang weather station, which is closest to the study region.

### 2.3. Calculation and Gradation of FVC

In this paper, the FVC (fractional vegetation coverage) in the study region was calculated using the pixel binary model. It was assumed that pixel information $S$ is composed of land covered by vegetation and bare land not covered by vegetation. If the remote sensing information of the pixel of the land covered by vegetation is $S_{veg}$, and that of the bare land is $S_{soil}$, the vegetation coverage $f_c$ could be expressed as [25]:

$$f_c = (S - S_{soil})/(S_{veg} - S_{soil}) \tag{1}$$

The *NDVI* has a close correlation with vegetation coverage [26]. The *NDVI* of pixels was used to calculate the FVC through the following formula:

$$f_c = (NDVI - NDVI_{soil})/(NDVI_{veg} - NDVI_{soil}) \tag{2}$$

where $NDVI_{soil}$ represents the *NDVI* of bare land, of which the value is close to 0 and does not change with time in theory. Furthermore, $NDVI_{soil}$ is not a fixed value due to influences from such factors as atmospheric conditions, solar radiation, and surface conditions. $NDVI_{veg}$ refers to the NDVI of land covered by vegetation, which is also affected by factors, such as vegetation type and seasonal change. Therefore, $NDVI_{soil}$ and $NDVI_{veg}$ are commonly replaced with $NDVI_{min}$ and $NDVI_{max}$, and thus, the following calculation formula was obtained:

$$f_c = (NDVI - NDVI_{min})/(NDVI_{max} - NDVI_{min}) \tag{3}$$

Due to noise in the images, the $NDVI_{min}$ and $NDVI_{max}$ cannot be selected directly from images, so the appropriate NDVI values should be selected as $NDVI_{min}$ and $NDVI_{max}$ according to the actual situation. In this study, the cumulative frequency was calculated based on the NDVI frequency distribution histogram in the study region for different periods. Finally, the NDVI values at 5% and 99% of the cumulative frequency were determined as $NDVI_{min}$ and $NDVI_{max}$, respectively.

The FVC in the study region was divided into four grades (Table 1) according to the Fourth National Desertification and Sandification Monitoring Technical Regulations of

the State Forestry Administration, the Standards for Classification and Gradation of Soil Erosion of the Ministry of Water Resources, and the standards for gradation of vegetation coverage in arid areas as well as the actual conditions of the study region [27].

**Table 1.** FVC types in the Taitema Lake region.

| Grade | Vegetation Classification Types | FVC % |
|-------|--------------------------------|-------|
| I | Very Low Coverage of Vegetation | <30 |
| II | Low Coverage of Vegetation | 30–50 |
| III | Medium-high Coverage of Vegetation | 50–70 |
| IV | High Coverage of Vegetation | 70–100 |

### 2.4. Dynamic Index and Trend Analysis

The dynamic index, which describes the degree of quantitative change of a certain type of vegetation coverage in a study region for a specific time period, was calculated through the following formula:

$$R_s = \frac{(U_b - U_a)}{U_a T} \times 100\% \tag{4}$$

where $R_s$ is the dynamic index of vegetation coverage, $U_a$ and $U_b$ indicate the number of a certain type of vegetation coverage type at the beginning and end of the study period, and $T$ refers to the length of the study period, in years. To reflect the variation trend of vegetation coverage $f_c$ in the study region in a period of time, the least square linear fitting change rate of vegetation coverage was calculated through the following formula [28,29]:

$$\theta_{slope} = \frac{n \sum_{i=1}^{n}(i \times f_{ci}) - \sum_{i=1}^{n} i \times \sum_{i=1}^{n} f_{ci}}{n \sum_{i=1}^{n} i^2 - \left(\sum_{i=1}^{n} i\right)^2} \tag{5}$$

where $\theta_{slope}$ represents the variation trend slope of vegetation coverage in each pixel, $n$ is the cumulative number of years in the study period, $i$ refers to the sequence of years ($i = 1, 2, \ldots, n$), and $f_{ci}$ is the vegetation coverage value of the $i$th year. Using this formulation, $\theta_{slope} > 0$ means the vegetation coverage is on the rise with time, while $\theta_{slope} = 0$ indicates that the vegetation coverage does not change with time, and $\theta_{slope} < 0$ suggests that the vegetation coverage has a decreasing trend with time. The larger the absolute value of $\theta_{slope}$, the more drastic the variation in vegetation coverage will be.

### 2.5. Coefficient of Variation

The coefficient of variation indicates the degree of dispersion of a set of data. The fluctuation of vegetation coverage can reflect the ecosystem stability in a region [30] and was calculated through the following formula:

$$CV_{f_c} = \frac{\sqrt{\frac{1}{(n-1)} \sum_{i=1}^{n} \left(f_{ci} - \overline{f_c}\right)^2}}{\overline{f_c}} \tag{6}$$

where $CV_{f_c}$ is the coefficient of variation of vegetation coverage, $n$ is the cumulative number of years in the study period, $i$ refers to the sequence of years ($i = 1, 2, \ldots, n$), $f_{ci}$ is the vegetation coverage value of the $i$th year, and $\overline{f_c}$ represents the average vegetation coverage during the study period. A larger value of $CV_{f_c}$ suggests that the ecological environment in the region is more fragile and vulnerable to environmental fluctuations.

### 2.6. Pixel-Based Hurst Index and Future Trend Analysis

The Hurst index was developed by the British hydrologist Harold Edwin Hurst to determine whether there is a long-term memory change trend in time series data. It can be

used to predict the future development and change trend of spatiotemporal series. Currently, the Hurst index can be estimated through several different methods, but the rescaled range (R/S) analysis is more reliable than other methods for estimation purposes [31]. In this paper, the Hurst index of vegetation coverage in the study region was estimated through the R/S analysis, and the time series $f_{ci}$ ($i$ = 1, 2, 3, ..., $n$) of vegetation coverage ($f_c$) had the following definition for any positive integer m:

1. Difference sequence:

$$\Delta f_{ci} = f_{ci} - f_{ci-1} \tag{7}$$

2. Mean sequence:

$$\overline{\Delta f_c(m)} = \frac{1}{m} \sum_{i=1}^{m} \Delta f_{ci} \tag{8}$$

3. Cumulative deviation:

$$X(t) = \sum_{i=1}^{m} \left( \Delta f_{ci} - \overline{\Delta f_c(m)} \right) \tag{9}$$

($1 \le t \le$ m).

4. Range:

$$R(m) = \max X(t) - \min X(t) \tag{10}$$

($1 \le$ m $\le$ n).

5. Standard deviation:

$$S(m) = \sqrt{\frac{1}{m} \sum_{i=1}^{m} \left( \Delta f_{ci} - \overline{\Delta f_c(m)} \right)^2} \tag{11}$$

m = 1, 2, 3, ... , n.

R, S and m meet $R(m)/S(m) \propto m^H$, i.e., $R(m)/S(m) = cm^H$, where $c$ is a constant, $R(m)/S(m)$ is the rescaled range, and H is the Hurst index, implying that the time series has the Hurst phenomenon. The logarithm of values on both sides of the equal sign in formula (11) was taken:

$$\log(R/S)_m = \log b + H \log m \tag{12}$$

Next, the Hurst index was obtained through least squares fitting with $\log m$ sequence in formula (12) as the independent variable and $\log(R/S)_m$ sequence as the dependent variable.

$H$ = 0.5 indicates that there is no long-term correlation in the time series, and the vegetation coverage ($f_c$) will change randomly in the future. $0.5 < H < 1$ suggests that the time series changes constantly, and the future variation trend of vegetation coverage will be consistent with that in the past. $0 < H < 0.5$ implies that the time series changes inversely, and the future variation trend of vegetation coverage will be opposite to that in the past. The closer the value of $H$ is to 0 and 1, the stronger the anti-continuity and continuity will be. The roadmap of this paper is as follows (see Figure 2).

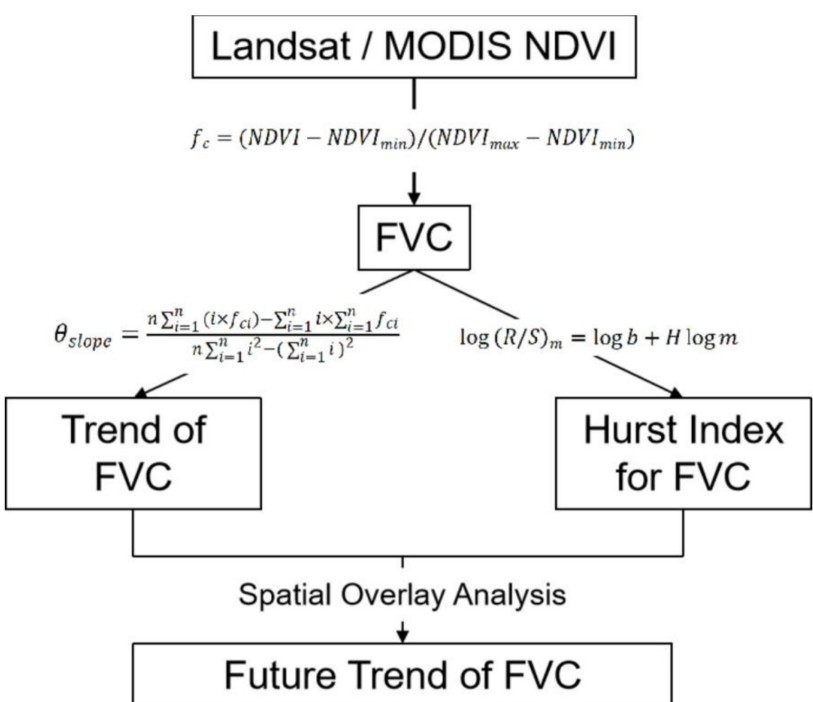

**Figure 2.** The Flowchart of methodology.

## 3. Results and Analyses

### 3.1. Spatial and Temporal Variations of Vegetation Coverage

The areas of different FVC types for four periods were calculated as per the FVC grading standard defined in the previous text (Tables 1 and 2). According to Figure 3 (the spatial distribution of FVC in the Taitema Lake region in 2001, 2007, 2014, and 2018) and Figure 4 (the changes and dynamic degrees of areas of different FVC types during the four periods), it can be seen that FVC in the study region showed a positive trend in space. In particular, the areas of low, medium-high, and high FVC near the lake entrance channel of the Tarim River expanded significantly. The areas of extremely low FVC accounted for 99.75%, 95.30%, 91.10%, and 64.87% in the four periods, respectively, which were the largest and gradually declined. The area ratios of low, medium-high, and high FVC showed an increasing trend (Table 2).

**Table 2.** Vegetation coverage areas of different classes in 2001, 2007, 2014, and 2018.

| Vegetation Types | 2001 | | 2007 | | 2014 | | 2018 | |
|---|---|---|---|---|---|---|---|---|
| | km$^2$ | % | km$^2$ | % | km$^2$ | % | km$^2$ | % |
| I | 510.60 | 99.75 | 487.82 | 95.30 | 466.32 | 91.10 | 332.06 | 64.87 |
| II | 1.07 | 0.21 | 15.77 | 3.08 | 22.57 | 4.41 | 111.74 | 21.83 |
| III | 0.15 | 0.03 | 5.02 | 0.98 | 11.93 | 2.33 | 41.72 | 8.15 |
| IV | 0.06 | 0.00 | 3.28 | 0.64 | 11.06 | 2.16 | 26.41 | 5.16 |

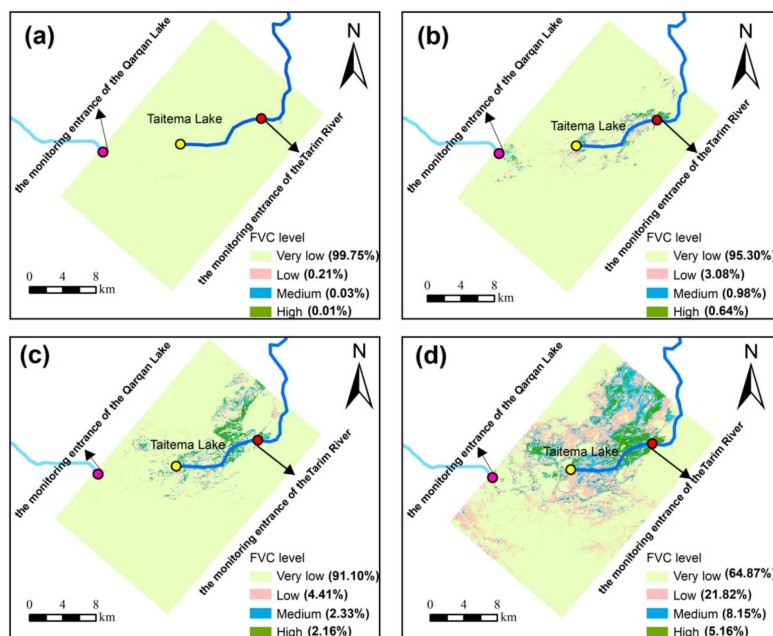

**Figure 3.** Spatial distributions of the fractional vegetation cover of the Taitema lake region in 2001 (**a**), 2007 (**b**), 2014 (**c**), and 2018 (**d**).

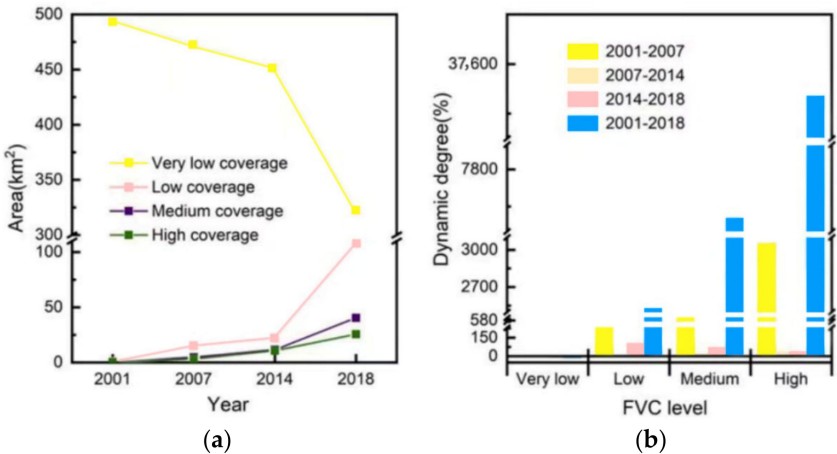

**Figure 4.** Area changes of different fractional vegetation cover types in Taitema lake region (**a**) and dynamic degree (**b**).

Furthermore, in the two periods of 2001–2007 and 2007–2014, the areas of extremely low FVC showed no significant reduction, and the dynamic degree was −0.74% and −0.63%, respectively. In 2001–2007, the areas of low, medium-high, and high FVC were increased significantly, with the most dramatic increase in the high FVC area, and the dynamic degree was 222.98%, 580.79%, and 3051.75%, respectively. In 2007–2014 and 2014–2018, there were obviously larger areas of low, medium-high, and high FVC, with increases of 6.58 km$^2$, 6.68 km$^2$, and 7.57 km$^2$ and 86.34 km$^2$, 28.83 km$^2$, and 14.84 km$^2$, respectively. The dynamic degree was moderate, which was 6.15%, 19.59%, and 34.36% as well as 98.79%, 62.39%, and 34.60%, respectively.

Compared with 2007–2014, the changes were more dramatic in 2014–2018. On the whole, the extremely low FVC area in the Taitema Lake region was decreased significantly after the implementation of the ecological water conveyance project in 2000, and the low, medium-high, and high FVC areas rose significantly. In particular, the areas of medium-high and high FVC expanded rapidly, showing a dynamic degree of 7403.97% and 37,338.16%, respectively. This is because there were almost no areas of medium-high

and high FVC around 2000. These data suggest that ecological water conveyance provides sufficient water sources for the study region and is important for improving FVC in the Taitema Lake region.

The linear trend of FVC dynamics in the study region was calculated based on Formula (5). Figure 5 displays the spatial trend of FVC dynamics in the Taitema Lake region in 2000–2018. It was found that FVC increase was observed in 24.02% of the study region, with a significant increase ($p < 0.05$) in 9.71% and a non-significant increase in 14.31% of the study region. The increases were mainly concentrated in the northeast of Taitema Lake, namely around the lake entrance channel of the Tarim River. This is because with the implementation of EWCP in the lower reaches of the Tarim River, there are sufficient water sources providing favorable conditions for vegetation growth, enabling a gradual rise in the FVC in the region. A decline in FVC, however, was found in 75.98% of the study region, with 15.78% having a significant decline ($p < 0.05$) and 60.20% having a non-significant decline. After the implementation of EWCP, the water area was raised to 463.79 km$^2$ in 2018 from 49 km$^2$ prior to the implementation of the water conveyance.

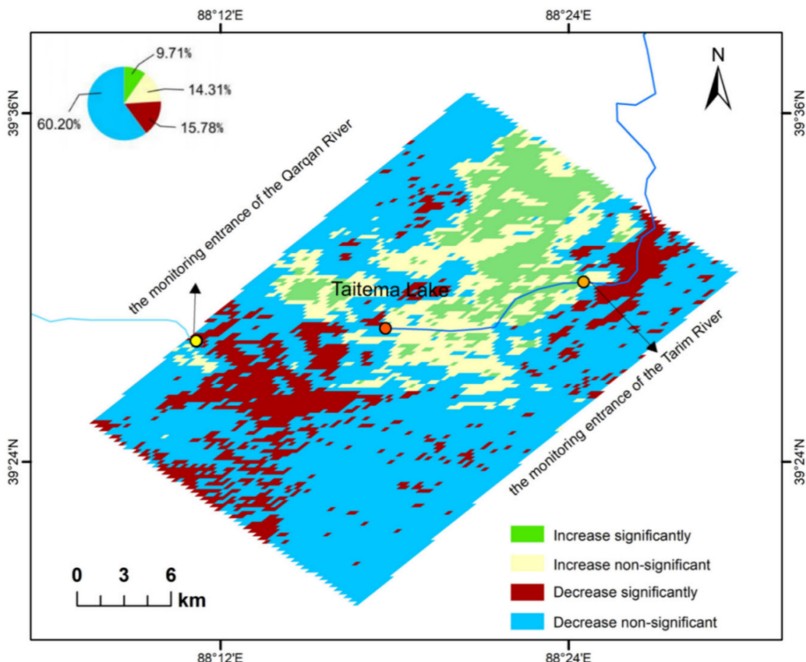

**Figure 5.** Spatial variation trend of fractional vegetation cover of the Taitema lake region from 2000 to 2018.

On the other hand, the rapidly enlarged seasonal water area of the lake has led to drowned vegetation in regions with extremely low coverage, shrinking the FVC area. Non-significant decline was found in regions far away from the Tarim River channel and the lake entrance of the Qarqan River. These areas were located far from the center water level of the water conveyance to the lake, showing that the groundwater level uplift did not radiate to the edge region. The vegetation growth in these regions mainly depends on groundwater, as surface water conveyance is seasonal, hindering seed fall and plant germination and growth.

It is also worth noting that the existing water conveyance along the river channel cannot expand and increase surface FVC or resist wind and stabilize sand due to the strong evapotranspiration in arid regions, the influx of sand blown by the wind, and the impact of groundwater quality. The vegetation growth in these regions is less affected by rapid river water replenishment due to ecological water conveyance. However, it fluctuates slightly with groundwater conditions, manifesting as either a slight increasing or decreasing trend. A significant FVC decrease was mainly discovered in regions with extremely low FVC. Since the implementation of EWCP, however, the area of extremely low FVC in the study

region has been reduced greatly on the whole, while areas of low, medium-high, and high FVC have been substantially enlarged.

### 3.2. Features of Variation in Fractional Vegetation Coverage

#### 3.2.1. Coefficient of Variation of Fractional Vegetation Coverage

Large fluctuations caused by human factors are found in ecological water conveyance. Hence, in this study, the variation coefficient of FVC in the Taitema Lake region in 2000–2018 (Figure 6, Table 3) was calculated based on variation coefficient (Equation (6)) pixels and analyzed to explore the stability of vegetation restoration after the introduction of ecological water conveyance. The results showed that the stability of vegetation coverage in the study region gradually increased from the inside to the outside. A high fluctuation in vegetation coverage was found in only 0.16% of the study region, while relatively high and medium fluctuations were observed in 1.80% and 4.95%, mainly in the middle and northeast, respectively. Relatively low fluctuation was detected in 25.91% of the study region at the periphery of the areas with relatively high and medium fluctuations, while low fluctuation was found in 67.18%.

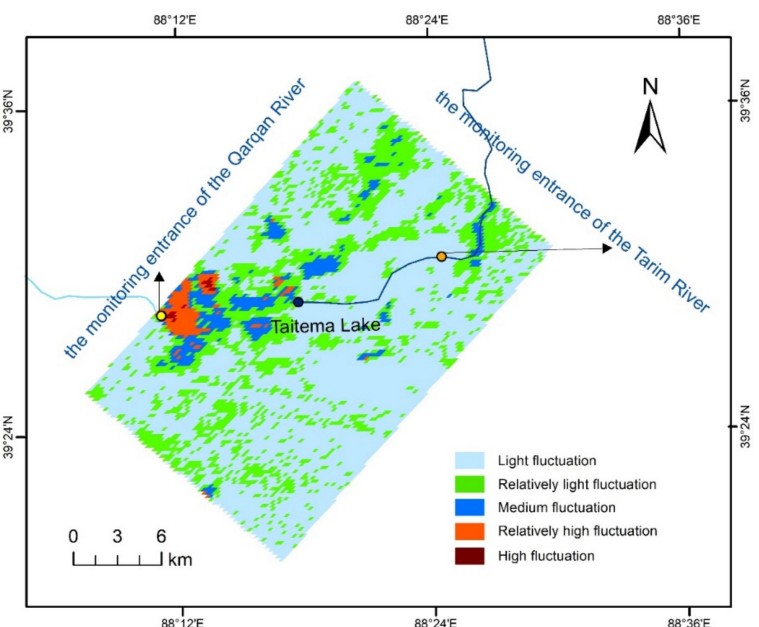

**Figure 6.** Variation coefficient of the fractional vegetation cover in Taitema lake region.

**Table 3.** Statistics of variation coefficient of fractional vegetation coverage in Taitema lake region.

| $CV_{FVC}$ | Variation Coefficient | % |
|---|---|---|
| $CV_{FVC} \geq 1.50$ | High Fluctuation | 0.16 |
| $0.85 \leq CV_{FVC} \leq 1.50$ | Relatively High Fluctuation | 1.80 |
| $0.65 \leq CV_{FVC} \leq 0.85$ | Medium Fluctuation | 4.95 |
| $0.45 \leq CV_{FVC} \leq 0.65$ | Relatively Light Fluctuation | 25.91 |
| $CV_{FVC} \leq 0.45$ | Light Fluctuation | 67.18 |

The growth status of vegetation in Taitema Lake mainly depends on water conditions. The study region can be divided into three types of areas according to these conditions. The first type is areas with stable and sufficient water conditions, including around the lake entrance channel of the Tarim River (i.e., the northeast). This area saw a slight fluctuation in vegetation coverage. The second type of area has fluctuating and abundant water sources and mainly includes the middle part of the study region. It is situated at the end of the ecological water conveyance and the Qarqan River, so it is difficult for the water to reach this area if there is little water in the river. As a result, the vegetation coverage

in this area had medium-high fluctuations. The last type of area is described as being relatively stable but having insufficient water conditions. It is located in the southeast and on the periphery of the study region. The vegetation coverage there is extremely low, primarily due to insufficient water. Further, the vegetation growth mainly depends on precipitation and groundwater, so the vegetation is sparse but relatively stable and exhibiting low fluctuations.

### 3.2.2. Hurst Index and the Future FVC Variation Trends

Based on the R/S analysis method, the Hurst index of FVC in the Taitema Lake region from 2000 to 2018 was estimated by pixel, and the Hurst index spatial distribution chart for FVC in the Taitema Lake region was obtained (Figure 7a). Recent levels of water conveyance show that the average Hurst index of FVC in the Taitema Lake region was 0.504, indicating that the future trend of FVC dynamics in that region will be consistent with its dynamics in the past. The portion of the study area with Hurst > 0.5 was 68.57%, meaning that FVC presents a trend of continuous improvement or degradation. Moreover, 0.58% of the region showed strong continuity (0.75 < Hurst), while 67.99% of the region showed relatively poor continuity (0.5 < Hurst < 0.75). The remaining 31.43% of the study region had discontinuous changes (0 < Hurst < 0.5). However, discontinuous changes would be shown in future FVC rather than past. Spatially, the regions with continuous changes were mainly located on the periphery of the study area and the outer layer of the river channels, while the regions with discontinuous changes were mainly located in the middle part of the lake or around the lake entrance of the river channel.

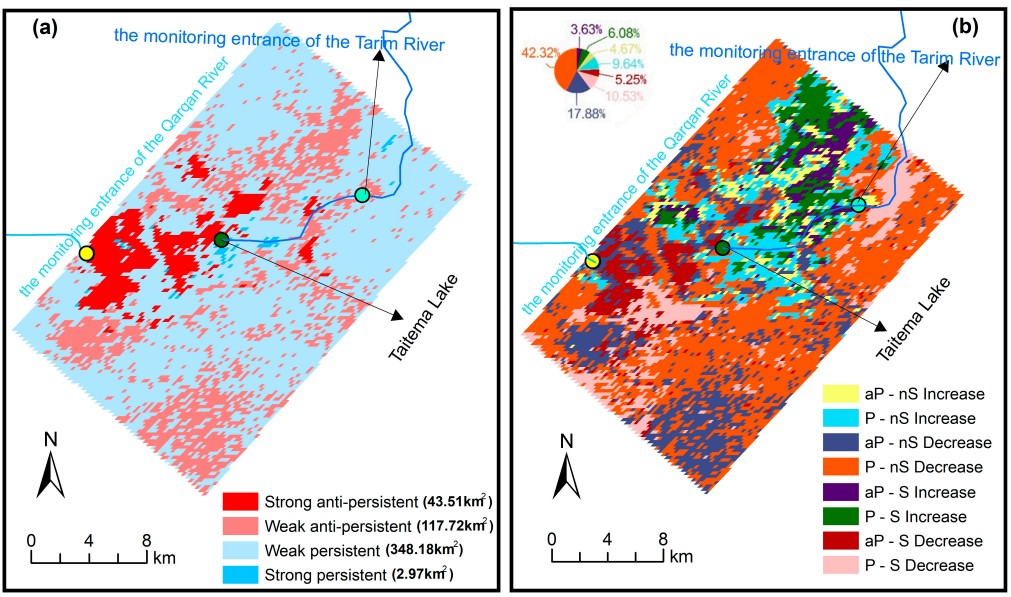

**Figure 7.** Hurst exponent of the fractional vegetation cover (**a**) and future trends (**b**) in Taitema lake region (ap: anti-persistent; p: persistent; S: Significantly; nS: non-Significantly).

To further predict the spatial features of FVC dynamics in the study region, overlay analysis was carried out on the FVC dynamics trend chart and the Hurst index spatial distribution chart. The continuity distribution chart for FVC dynamics trends in the Taitema Lake region from 2000 to 2018 (Figure 7b) was then obtained and divided into eight scenarios. After analysis, it can be seen that the regions with continuous and discontinuous changes of non-significant increase and those with continuous and discontinuous changes of non-significant decrease accounted for 9.64%, 4.67%, 42.34%, and 17.88%, respectively. This indicates that 74.51% of the study region will not have significant variations in FVC in the future. Furthermore, the proportions of regions with continuous changes of significant increase and discontinuous changes of significant decrease were 6.08% and 5.25%, respec-

tively, and were mainly located in the middle part of the study region and surrounding regions of the lake entrance channel of the Tarim River.

It is anticipated that 11.33% of the study region will experience a significant increase in FVC in the future. The regions with continuous changes of significant decrease and discontinuous changes of significant increase comprised 10.53% and 3.63%, respectively, mainly where FVC was extremely low or low. This suggests that these areas will likely continue to reduce in the future because as the ecological water conveyance continues, the groundwater level will be elevated year by year to produce a cumulative effect. The outcome of this dynamic is that regions with low FVC that are mainly affected by the groundwater level will gradually decrease.

Overall, given the current level of water conveyance in the study region, the FVC dynamics in the future are generally consistent with those of the past. This means that 74.51% of the Taitema Lake region would have no obvious variation trend in FVC with few fluctuations, 11.33% of regions with medium and high FVC would expand, and 14.16% of regions with extremely low and low FVC would shrink.

## 4. Discussion

### 4.1. Major Factors Influencing Temporal and Spatial Variations of Water Volume in the Taitema Lake

The Taitema Lake region was divided into four different zones (A, B, C, and D; see Figure 8) based on our studies in combination with other factors. These factors include remote sensing images, lake distribution pattern, nourishment source, lake region elevation, and location of the entrance before and after the diversion of the Qarqan River in 2002. After the completion of the diversion [32], the new river channel will divert the water into zone D.

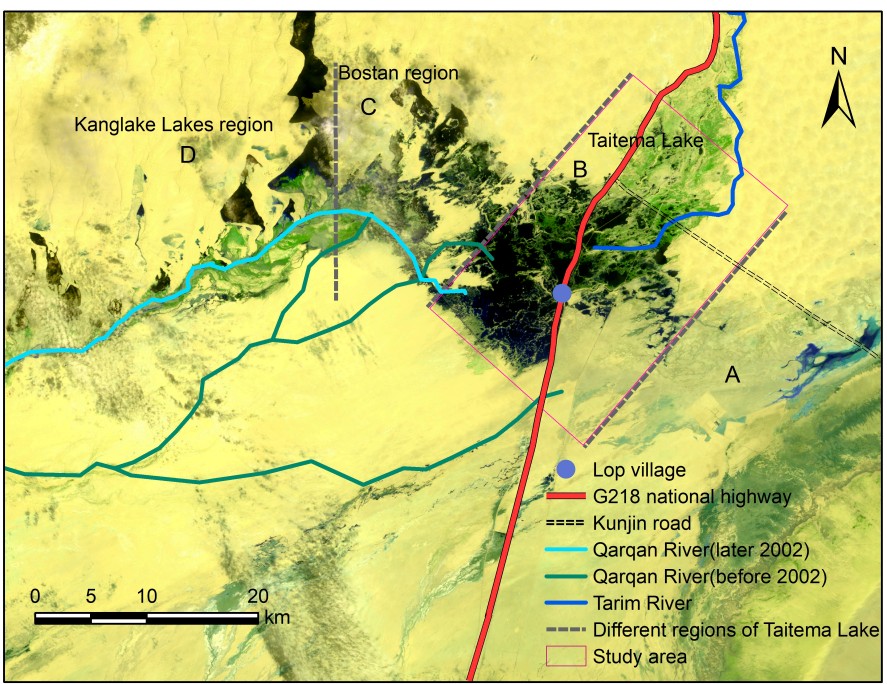

**Figure 8.** Sketch map of different zones in the study area.

In the case of sufficient water inflow, part of the water flowing through zone D will enter zone C (namely the Bostan region). In zones C and D, with a mean elevation of 830 m, a group of Kanglake lakes have been formed. Some of the water flows through zone C along the river channel and enters zone B, which has a natural boundary separating it from zone C. Zone B is the Taitema Lake region where the Tarim River (elevation 801.5 m) flows into it. This shows that the water from the Tarim River can barely flow into zone C. It is

also worth noting that the National Highway 218 passes through zone B and connects with the Kunjin Highway. When a great amount of water comes from the Tarim River, the lake water in zone B flows to zone A, east of National Highway 218. A lake could be formed there when enough inflow water is present. In this paper, the FVC in zone B was mainly studied using the natural division between zones B and C as the boundary.

The water conveyance amount to the lower reaches of the Tarim River mainly relies on the amount of water from the upper reaches of the river. Temporally, the ecological water conveyance amount was generally on the rise from 2000 to 2018 (Figure 8), increasing from $9.92 \times 10^7$ m$^3$ in 2000 to $2.83 \times 10^8$ m$^3$ in 2005 and reaching $7.00 \times 10^8$ m$^3$ in 2018. However, the water conveyance amount decreased and even reached zero in the middle years, displaying the trend of increasing initially and decreasing afterwards. Thus, the water conveyance amount was increased to $2.83 \times 10^8$ m$^3$ from 2000 to 2005, gradually decreased to 0 m$^3$ in 2008, and then gradually rose again starting in 2009, reaching $8.52 \times 10^8$ m$^3$ in 2011, and falling to $4.61 \times 10^8$ m$^3$ in 2015. Thereafter, it rose from $6.76 \times 10^8$ in 2016 m$^3$ to $7.00 \times 10^8$ m$^3$ in 2018 [14]. Hence, the overall trend was increasing.

During the 19 years of ecological water conveyance, the origins of the Tarim River (Hetian River, Yarkant River, and Aksu River) could not supply a normal water conveyance amount in 2000–2009. In other words, the volume of the water discharged from the three origins at the upper reaches of the Tarim River to the Daxihaizi Reservoir was still smaller than the average planned water conveyance amount to the lower reaches of the Tarim River. As a result, the runoff of the Alaer (the main stream of the Tarim River) declined to $28.57 \times 10^8$ m$^3$, $24.74 \times 10^8$ m$^3$, and $14.80 \times 10^8$ m$^3$ in 2007, 2008, and 2009, respectively. The water conveyance line from the Daxihaizi Reservoir to the Taitema Lake was 321-km long, and the immediate start of emergency ecological water conveyance from Bosten Lake was still unable to meet the demand of downstream ecological water consumption. In 2000, 2006, 2007, and 2009, the downstream waterhead could not enter the Taitema Lake [14], and the ecological water conveyance was even stopped in 2008. Taitema Lake was dry until the spring of 2010 [32], with crescent dunes forming at the bottom of the lake due to wind erosion, reaching mild desertification.

Given this situation, major measures were taken, such as strengthening the ecological management in the upper reaches of the Tarim River and reducing water consumption at the three origins in the Taitema River Basin. The result was that plentiful water flowed from the upper reaches, the waterhead entered the Taitema Lake every year, the surface of the Taitema Lake gradually expanded, and the groundwater level was elevated in 2010–2016. However, in 2014, the lake surface shrank sharply, mainly caused by the decreased amount of water entering the lake due to the closure of some ecological gates between the Daxihaizi Reservoir and Taitema Lake. Water conveyance was then conducted throughout the Tarim River Basin in 2017–2018, and the water surface of Taitema Lake was greatly enlarged, even reaching 463.79 km$^2$ for a time [20]. As a result, the Tarim River Basin became a green ecological barrier for wind prevention and sand fixation. Ecological water conveyance is mainly controlled by humans, and human-controlled ecological water conveyance is also affected by extreme climates and regional water-consumption changes. The uncertainty of ecological water conveyance directly gives rise to fluctuations in the water volume of Taitema Lake.

## 4.2. Spatial Aggregation Trend of FVC

The results of this study uncovered that the FVC in the Taitema Lake region presents a positive development on the whole since the start of the water conveyance, with significant increases in low, medium-high, and high FVC areas and an unexpected increase in FVC (Figure 9). These changes demonstrate the remarkable role of ecological water conveyance in the promotion of vegetation growth in the study region.

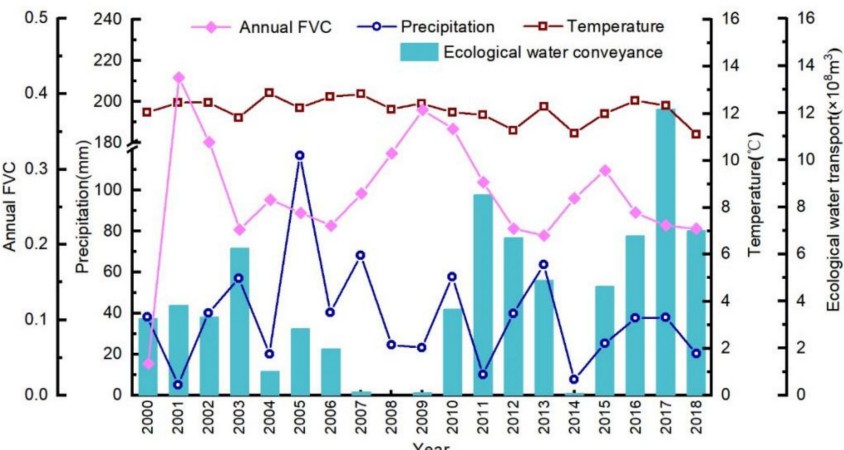

**Figure 9.** Changes in mean annual FVC, ecologic water conveyance, average annual temperature, and annual precipitation (Ruoqiang station) in the study area.

In addition, all the regions with increased FVC were surrounding the lake entrance channel of the Tarim River and the lake entrance of the Qarqan River, with greater increases occurring the less the distance was to the river channel and lake entrance. Similarly, the regions closer to the river channel would have more stable water conditions and smaller fluctuations in FVC. It was also found that the fluctuations were negligible in regions with extremely low FVC. The reason for the slightness of the fluctuations is twofold: on one hand, these areas are far from the river channel or the lake entrance of the river channel, so it is difficult for them to obtain the ecological water or the water supply from the Qarqan River; on the other hand, the original water conditions in these regions were dismal, leaving the growth of vegetation to rely mainly on the scarce groundwater and precipitation. Therefore, the regions' extremely low FVC remained relatively stable for a long time.

In the future, the restoration of riverbank vegetation is feasible through ecological water conveyance at the existing level [33]. However, since the water-receiving area of the ecological water conveyance is limited, the regions with notably increased FVC would likely also be located near the lake entrance channel of the Tarim River. Nonetheless, future FVC near the lake entrance of the Qarqan River would probably experience a notable increase. This is because with the progress of ecological water conveyance, a large amount of water flows into the Taitema Lake region every year, which effectively relieves the supply of the Qarqan River run-off to the groundwater around and allows more water to flow into Taitema Lake. With the elevation of the groundwater level and the increase in the Qarqan River run-off into the lake, sufficient and stable amounts of water will flow into the middle part of Taitema Lake to satisfy the growth of vegetation. Consequently, future FVC in this area would chart an increasing trend.

### 4.3. Water Volume of Taitema Lake, a Crucial Factor Affecting FVC and Ecological Restoration

Temperature and water are vital environmental factors influencing vegetation growth. The latter is especially critical. Since the start of the ecological water conveyance project, the water resources for the vegetation in the Taitema Lake region mainly have close associations with precipitation and groundwater replenishment, along with ecological water conveyance. Since the implementation of EWCP for the Tarim River in 2000, the ecological water conveyance has greatly affected the water area of Taitema Lake. For instance, the surface wetland area expanded from 49.00 km² prior to the start of the project to 463.79 km² in 2018 [20]. Taitema Lake has "revived" [14], and FVC in the lake region continues to increase, consistent with the results of previous studies [34–36]. These results render a scientific basis for the rise of water level in Taitema Lake and the overall improvement of the basin's ecological environment.

Groundwater is a vital water resource for the survival of desert vegetation in arid regions such as the Tarim Basin [37], and its replenishment is increased by the ecologi-

cal water conveyance in the lower reaches of the Tarim River. The spatial difference of groundwater replenishment in arid regions mainly lies in the distribution of surface water and its linear replenishment from rivers as well as in the water supply to the irrigation areas. Approximately 90% of the total annual average groundwater replenishment comes from surface water, while the effect of rainfall infiltration on groundwater replenishment is weak [37,38]. The groundwater in the middle and lower reaches of the Tarim River is mainly replenished by surface river infiltration [39].

The annual average temperature and annual precipitation recorded at the Ruoqiang weather station, which is the closest weather station to the study region, were analyzed. The analysis indicated that both temperature and precipitation exhibited a very mild decreasing trend (Figure 9). Precipitation contributed to the growth of vegetation to a certain extent, but its slight decrease in the study region did not cause any negative influence on FVC. These findings suggest that in the current period, ecological water conveyance remains the primary factor influencing FVC in the study region and also the primary source of groundwater replenishment in the Taitema Lake region [14].

With the continuous implementation of EWCP, the groundwater level in the lower reaches of the Tarim River has increased [13,14,16,17,19], and the ecological water level has held steady at 4–6 m. The water is most abundant when the groundwater level is 6 m in arid regions, and trees, such as Populus euphratica and Tamarix ramosissima, grow well and thrive. When the groundwater level is 0–4 m, obvious niche overlap of trees, shrubs, and herbs is observed [40]. However, some deep-rooted shrubs cannot obtain sufficient water when the groundwater level is lower than 4 m and so whither and deteriorate [41]. The lowest indexes, including the vegetation diversity index, the Shannon–Wiener index, and species richness, are clearly detected if the groundwater level is less than 9 m [42].

Since the implementation of the ecological water conveyance, 26 species of live plants have been found in the Taitema Lake region [43]. As well, the ecology of local regions has begun to recover and improve, as the purpose of the ecological water conveyance is to replenish and uplift the groundwater level in the lake region. Due to fluctuations in groundwater levels, there is a difference in the growth of different types of vegetation across various zones of the Taitema Lake region. Furthermore, because groundwater level is closely related to the diversity of population and plant growth, the amount of ecological water conveyance must reach a certain value to stabilize the water volume in the lake region, as only constant water replenishment can maintain the groundwater level for certain plant species.

As EWCP progresses, the Taitema Lake region has gradually expanded, and the groundwater level has risen to an ecological water level suitable for plant species. This has facilitated the continuous increase of FVC, elevating the number of species and preventing the closure of deserts on both sides of the lake region. As a result, the ecological environment of the lake region is effectively improved, and the ecosystem is gradually recovering [44]. However, these improvements are the result of the continuous progress of EWCP; reverse outcomes may be obtained if EWCP is not implemented or cannot be implemented in time [17]. As shown in the survey, the water volume of the terminal lake declined in 2007–2009 due to a decrease in the runoff of the Tarim River induced by climate changes and the reduced conveyance of ecological water to the lower reaches of the river caused by excessive water consumption in the oases at the upper reaches. At the same time, Taitema Lake started drying up again, the deserts on both sides advanced to the lake region, the groundwater level of the basin dropped, the FVC was reduced, and the ecological environment deteriorated. Considering how quickly the improvements can be nullified, the management of regional water resources for production and livelihood should be combined with the ecological water demand in the lower reaches of the basin so as to effectively guarantee and enhance the current and future economic and ecological benefits from EWCP.

## 5. Conclusions

Based on Landsat TM/OLI remote sensing images and MODIS data products, fractional vegetation coverage in the Taitema Lake region was studied in this paper following the implementation of the ecological water conveyance project in 2000. Specifically, the FVC was extracted and calculated, and its characteristics and future trend dynamics for the study area were analyzed and predicted through trend analysis and the pixel-based Hurst index. Additionally, the effect of the run-off from the Tarim River on the FVC in the Taitma Lake region was considered for the sake of comparison and the area divided according to the regional characteristics of the lake. The research scope of this paper was determined based on the restriction of the water area of the study region, which can be expanded in the next research to further study the ecological restoration process of the surrounding Kanglake Wetland Group.

Our conclusions for the present study are as follows:

(1) From 2000 to 2018, fractional vegetation coverage in the Taitema Lake region presented a positive trend on the whole following the implementation of the ecological water conveyance project. The extremely low FVC area decreased gradually but made up the largest proportion across various periods, and the areas of low, medium-high ,and high FVC increased gradually and were concentrated around the lake entrance channel of the Tarim River and the lake entrance of the Qarqan River, implying that the vegetation in the lake region has been well restored.

(2) After the start of the ecological water conveyance project, the area of regions with notably increased FVC around Taitema Lake accounted for 9.71% and were located near the lake entrance channel of the Tarim River. The area of regions with decreased FVC accounted for 75.98% and demonstrated extremely low FVC. The stability of FVC in the region weakened gradually from the periphery inwards. Approximately 93.09% of the region had relatively low and low fluctuations, and only 1.96% displayed relatively high and high fluctuations, mainly located in the middle part of the study region far from any water source. This is consistent with the distribution of water consumption and the restoration of groundwater level in various zones across the lake region.

(3) The average Hurst index of FVC in the Taitema Lake region since the start of the ecological water conveyance project was 0.504, with 68.57% of the region manifesting the same FVC dynamics trend in the future and 31.43% of the region exhibiting the opposite FVC dynamics trend. Additionally, 74.51% of the Taitema Lake region would likely show mild fluctuations in future FVC, 11.33% of the region would show an increasing trend mainly in the middle part of the lake or around the lake entrance channel of the Tarim River, and 14.16% of the region with extremely low and low FVC would show a sharply decreasing trend. In a scenario featuring continuous ecological water conveyance, future FVC in the study region would show a trend of ecological restoration.

(4) Compared to the influences of natural factors like temperature and precipitation on FVC in arid regions, the positive effect of ecological water conveyance was more obvious, so ecological water conveyance remains as the major route of ecological restoration for the Taitema Lake.

**Author Contributions:** Conceptualization, T.Z., Y.C. and J.D.; data curation, T.Z., Y.C. and J.D.; formal analysis, T.Z.; funding acquisition, Y.C.; investigation, T.Z.; methodology, T.Z.; supervision, Y.C. and J.D.; writing—original draft, T.Z., Y.C. and J.D.; writing—review and editing, T.Z. All authors have read and agreed to the published version of the manuscript.

**Funding:** This research was supported by the Natural Science Foundation of Xinjiang Uygur Autonomous Region, China (Grant No: 2021D01D01).

**Institutional Review Board Statement:** Not applicable.

**Informed Consent Statement:** Not applicable.

**Data Availability Statement:** Not applicable.

**Acknowledgments:** We are grateful to anonymous reviewers and editorial staff for their constructive and helpful suggestions.

**Conflicts of Interest:** The authors declare no conflict of interest.

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
