# Peer review of "Research on Vegetation Coverage Dynamics and Prediction in the Taitema Lake Region"

_water, doi:10.3390/w14050725_

Round 1

Reviewer 1 Report

Since I agreed to review this paper considering the title and its asbract, there are major flows regading the research methodology, as well as results elaboration and their presentation. In particular, I found it very difficult to understand authors' approach, and therefore evaluate their study.

First and morefost, I was unable to identify the research gap, along with the research methodolody, after reading the abstract of the paper. It is simply to long which greatly limits its readability and understanding. The same concerns introduction.

Secondly, the structure of the paper is invalid which again makes paper hardly understanable. Therefore, the paper is not well-organized. 

My overall impression is that, while the problem and reseach method seem to be promising, then the rest is simply a long and monotonous story. In other words, Authors failed to tell an interesting story regarding.

The current version of the paper should be reorganized and rewritten. Besides all maths formulas must be checked again.

Reviewer 2 Report

The article is interesting but it is more suitable for the Remote Sensing Journal as it has an almost insignificant connection to water issues except that related to previous studies (section 4.1). The methodology was applied to the lake vegetation.  Anyhow, it is the decision of the editor to publish it in Water J. or to transfer it to RSJ. The following comments could improve the quality of the paper.

1- Please start the introduction with a statement to show the importance of the topic globally then go for regional followed by local (case study).

2- Improve the methodology by adding a detailed flowchart (algorithm) to show all steps you followed to achieve the objectives/outcomes of the investigation.

3- Almost all the links embedded in the text are not valid. please rewrite the link and check that they are working properly. See for example links posted on pages 120 and 122.

4- It seems the last conclusion is not supported by the data/analysis presented in the paper. If it is a recommendation, show that and further recommendations for future studies, policy planners, and decision-makers. Also, please show how your study could help to achieve the investigated lake sustainability. 

5- It is advisable to add DOI to all references whenever it is possible.

Round 2

Reviewer 1 Report

I think that the current version of this manuscript can be considered for the publication since the Authors have addressed all the issues raised in my review.